

# Genome-wide identification and expression analyses of C2H2 zinc finger transcription factors in *Pleurotus ostreatus*

Qiangqiang Ding[1,2], Hongyuan Zhao[1], Peilei Zhu[1,2], Xiangting Jiang[1], Fan Nie[1,2] and Guoqing Li[1,2]

[1] Institute of Horticulture, Anhui Academy of Agricultural Sciences, Heifei, Anhui Province, China
[2] Key Laboratory of Genetic Improvement and Ecophysiology of Horticultural Crops, Heifei, Anhui Province, China

## ABSTRACT

The C2H2-type zinc finger proteins (C2H2-ZFPs) regulate various developmental processes and abiotic stress responses in eukaryotes. Yet, a comprehensive analysis of these transcription factors which could be used to find candidate genes related to the control the development and abiotic stress tolerance has not been performed in *Pleurotus ostreatus*. To fill this knowledge gap, 18 *C2H2-ZFs* were identified in the *P. ostreatus* genome. Phylogenetic analysis indicated that these proteins have dissimilar amino acid sequences. In addition, these proteins had variable protein characteristics, gene intron-exon structures, and motif compositions. The expression patterns of *PoC2H2-ZFs* in mycelia, primordia, and young and mature fruiting bodies were investigated using qRT-PCR. The expression of some *PoC2H2-ZFs* is regulated by auxin and cytokinin. Moreover, members of *PoC2H2-ZFs* expression levels are changed dramatically under heat and cold stress, suggesting that these genes may participate in abiotic stress responses. These findings could be used to study the role of *P. ostreatus*-derived *C2H2-ZFs* in development and stress tolerance.

## INTRODUCTION

*Pleurotus ostreatus* is a mushroom that is widely cultivated for its nutritional value and relatively simple cultivation techniques (*Chang & Miles, 2004*; *Khan & Tania, 2012*). The production of *P. ostreatus* relies on the precise control of fruiting body development. The formation of fruiting bodies starts when two hyphae with different mating types combine to form dikaryotic hyphae during a process called plasmogamy. If these dikaryotic hyphae aggregate, they will develop into primordia, which will then differentiate into fruiting bodies. Genome sequencing of model mushroom *Schizophyllum commune* indicated that many predicted transcription factors like zinc finger proteins (ZFPs), MYB, fungal specific transcription factor (fst), and so on are differentially expressed during sexual development (*Ohm et al., 2010*). The *Pofst3* gene in *P. ostreatus* is a homolog of the *fst3* gene

Corresponding author
Guoqing Li, liguoqing1976@163.com

in *S. commune* and was determined to play a role in primordia formation (*Qi et al., 2019*). These results indicated that certain transcription factors may mediate the development of *P. ostreatus*. However, only a few transcription factors have been identified in this commercial mushroom. The identification and characterization of more transcription factors in *P. ostreatus* could help researchers identify interesting proteins involved in various development processes or help breeders selectively breed for controlled mushroom development.

ZFPs are one of the largest transcription factor families in eukaryotic genomes (*Laity, Lee & Wright, 2001*). The term "zinc finger" refers to proteins harbor a conserved domain consisting of cysteine (C) and/or histidine (H) residues. This domain binds with a zinc ion and, structurally, consists of a two-stranded antiparallel beta-sheet and a helix (*Takatsuji, 1998*). ZFPs can be divided into the following categories based on the number and location of C and H residues in this conserved domain: C2H2, C2HC, C2HC5, C2C2, C3H, C3HC4, C4, C4HC3, C6, and C8 (*Berg & Shi, 1996*). Among these, C2H2-type zinc finger proteins (C2H2-ZFPs) are the most widely studied. The zinc finger domain in these proteins contains two C and two H residues, which are described as $CX_{(2-4)}CX_{12}HX_{(3-5)}H$ (where X represents any amino acid) (*Pabo, Peisach & Grant, 2001*).

Functional analysis has shown that *C2H2-ZFs* participate in vegetative growth and reproductive development in plants (*An et al., 2012*; *Lu et al., 2012*; *Sun et al., 2015*). They also mediate growth and development (*Tian et al., 2017*), sexual development (*Kim et al., 2009*), oospores production (*Wang et al., 2009*), and so on in fungi and fungal hyphae. Moreover, while *c2h2*-overexpression strains did not affect normal development in *Agaricus bisporus*, the yield per day of the transgenic strains peaked 1 day earlier than the control strains did (*Pelkmans et al., 2016*). The effect of *C2H2-ZFs* on mushroom formation makes them a target for breeding of this commercial mushroom. However, no further studies have been conducted on the roles of *C2H2-ZFs* in *P. ostreatus* development thus far.

Apart from regulating various development processes, *C2H2-ZFs* have been found to play crucial roles in abiotic stress defense. In plants, they have been shown to respond to heat (*Mittler et al., 2006*), and functional analysis has shown they help temper the effects of drought (*Yin et al., 2017*), cold (*Liu et al., 2017*), and salt stress (*Ciftci-Yilmaz et al., 2007*). In China, traditional greenhouses are mainly used for cultivating *P. ostreatus*, but they often lack proper environmental control. Environmental stress, especially heat, consistently threatens the supply of greenhouse-grown mushrooms. Extremely and continuously high temperatures (>36 °C) disrupt the cell wall integrity of *P. ostreatus* and enhance the ability of *Trichoderma asperellum* to infect mycelia (*Qiu et al., 2018*). In addition, mycelia exposed to 40 °C for 3 h leads to the accumulation of lactate, which inhibits mycelial growth (*Yan et al., 2020*). On the other hand, treatment at 5 °C significantly decreases the activity of enzymes like laccase and Mn-peroxidase in mycelia (*Snajdr & Baldrian, 2007*). Therefore, identification of *C2H2-ZFs* in *P. ostreatus* could also be used to in breeding programs to improve environmental stress tolerance in mushrooms.

In this study, *C2H2-ZFs* in the *P. ostreatus* genome were identified and characterized using bioinformatic analysis. Then, the expression profiles of each transcription factor were measured in different tissues to better understand their roles in regulating *P. ostreatus*

development and stress response. The results of this work provide useful information about the characterization of *C2H2-ZFs* in mushrooms and candidate genes for the control the development and abiotic stress tolerance in *P. ostreatus*.

## MATERIALS & METHODS

### Identification and characterization of C2H2-ZFPs in *P. ostreatus*

A Hidden Markov Model (HMM) profile of the C2H2 domain sequences (PF00096) was downloaded from the Pfam database and used as a query in the HMMER3.0 program against the publicly available genome of *P. ostreatus* from JGI (http://genome. jgi.doe.gov/PleosPC15_2/) to search for C2H2-ZFPs with an *E*-value less than $1e^{-4}$. The candidate C2H2-ZFPs were submitted to SMART (http://smart.embl-heidelberg.de) to confirm the presence of a C2H2 domain. C2H2 domains that did not contain the "$CX_{(2-4)}CX_{12}HX_{(3-5)}H$" motif were deleted manually, and the rest were regarded as PoC2H2-ZFPs. The subcellular localizations of PoC2H2-ZFPs were predicted using WoLF PSORT (http://wolfpsort.org/). The ExPasy site (http://web.expasy.org/protparam/) was used to calculate the molecular weight (MW) and isoelectric point (pI) of the proteins.

### Phylogenetic analysis and multiple sequence alignment

A phylogenetic tree was constructed for C2H2-ZFPs in *P. ostreatus* using the MEGA-X program (*Sudhir et al., 2018*). A neighbor-joining (NJ) method based on the JTT model with bootstrapping was performed 1000 times to calculate phylogenetic distances.

Multiple sequence alignment was performed on full C2H2-ZFP and C2H2 domains using MEGA-X. The results were loaded into JaLview for visualization (*Waterhouse et al., 2009*).

### Gene structure and motif analysis

The exon-intron organization of the *C2H2-ZF* genes was obtained from genomic information and drawn using Tbtools (*Chen et al., 2020*). Then, the proteins were submitted to MEME (http://meme-suite.org/tools/meme) to identify conserved motifs with five motif numbers. The optimum motif length was fixed using the default parameters (6–50 residues).

### Strains, culture conditions, and sample collection

The *P. ostreatus 3125* strain was provided by the Institute of Scientific Edible Fungi, Gaoyou, China. The fungi were grown in potato dextrose agar (PDA) medium, then transferred to sterile wheat grain medium and cultured at 25 °C in the dark in a temperature-controlled incubator. Five days later, a bit of wheat grain and mycelium were placed into sterile growth bags composed of 60% cottonseed hulls, 35% corncob, 10% bran, 3% gypsum, and 2% kalk. They were cultured at 55% humidity, in the temperature-controlled incubator. The mycelium were collected once they were fully grown. To obtain primordia, the growth bags were transferred to the culture room (10–13 °C, 80% relative humidity). Young and mature fruiting bodies were collected on day 6 and 12, respectively, as primordia differentiated into fruiting bodies. All samples were frozen in liquid nitrogen and immediately stored at −80 °C.

## Hormonal and abiotic stress treatments

Selected primordia were exposed to hormones and environmental stresses and the response of *PoC2H2-ZFs* was analyzed. For the hormonal treatment, the primordia were covered with absorbent cotton soaked in 200 ul 0.01 mM IAA, 0.01 mM zeatin, and $H_2O$ then collected after 1 and 3 h (h). For the heat and cold stress treatments, the primordia were cultured at 38 °C and 4 °C in a temperature-controlled incubator for 1 and for 3 h, respectively. For the control, primordia were grown in the culture room at 10–13 °C.

## Isolation of RNA, cDNA synthesis, and qRT-PCR analysis

Total RNA was isolated from the sampled primordia using the Plant Total RNA Isolation Kit (Sangon Biotech Co., Ltd, ShangHai). cDNA was generated using the MightyScript First Strand cDNA Synthesis Master Mix (Sangon Biotech Co., Ltd, ShangHai) according to the manufacturer's protocol. The 2X SG Fast qPCR Master Mix (Sangon Biotech Co., Ltd, ShangHai) was used to perform qRT-PCR. The *sar* gene was used as the reference (*Castanera et al., 2015*) and the relative expression level of genes was analyzed using the $2^{-\Delta CT}$ or $2^{-\Delta\Delta CT}$ method. The primer sequences used for qRT-PCR are listed in Table S3. A heatmap showing relative expression levels of *PoC2H2-ZF* genes was generated using Tbtools (*Chen et al., 2020*).

# RESULTS

## Identification, characterization, and phylogenetic analysis of C2H2-ZFPs in *P. ostreatus*

Using an HMM and manual correction, 18 C2H2-ZFPs were identified in the *P. ostreatus* genome. All these proteins contained one to four conserved C2H2 domains either in the N-terminus or the C-terminus (Fig. S1). Detailed information about the characteristics of these proteins like amino acid size, MW, isoelectric points, and so on were also analyzed (Table S1). The results showed that the PoC2H2-ZFPs had between 149 and 688 amino acids, molecular weights ranging from 16.4 (*PleosPC15_2|1089905*) to 74.8 kDa (*PleosPC15_2|1054163*), and isoelectric points ranging from 4.66 (*PleosPC15_2|1079678*) to 10.8 (*PleosPC15_2|1089905*). All the PoC2H2-ZFPs are predicted to be nuclear proteins based on subcellular localization analysis (Table S1).

To analyze the phylogenetic relationships of these C2H2-ZFPs in *P. ostreatus*, a phylogenetic tree was constructed with their full protein sequences. As a result, the 18 PoC2H2-ZFPs were clustered into four separate clades (Fig. 1A). PleosPC15_2|1104202 is the only gene in clade III, indicating that it originated independently of the other genes. Based on bootstrapping values, the other genes are distantly related to each other. Most PoC2H2-ZFPs in the same clade had low bootstrap values (<60%). Only PleosPC15_2|1111338 and PleosPC15_2|1095114, which are regarded as duplicate genes, had strong bootstrap values (100%). The results of sequence alignment indicates that the PoC2H2-ZFPs share low sequence homology (Fig. S1).

## Gene structure and conserved motif analysis of *PoC2H2-ZFs*

To gain insights into the genetic structure of *PoC2H2-ZFs*, their exon-intron organization was analyzed. The results show that *PoC2H2-ZFs* have diverse gene structures and also

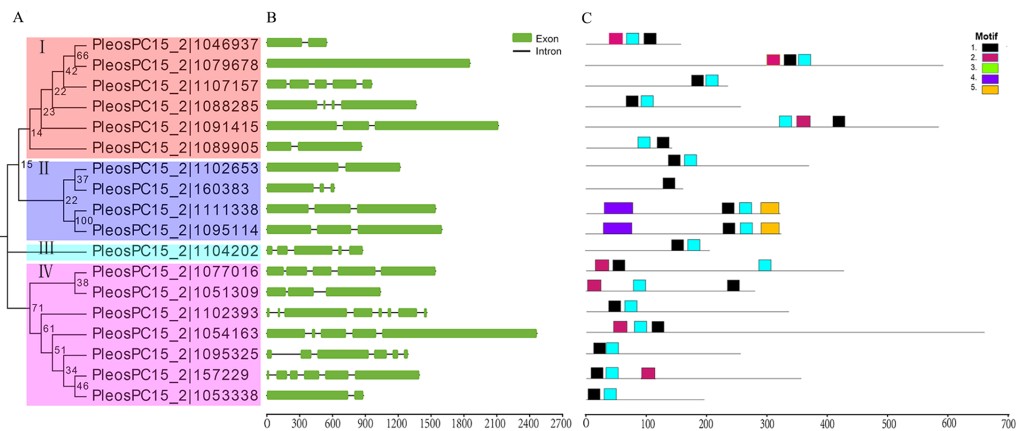

**Figure 1** **Phylogenetic relationships, gene structures, and conserved motifs analysis of *C2H2-ZFs* in *P. ostreatus*.** (A) The Phylogenetic tree of PoC2H2-ZFPs. The four major subfamilies are marked with different colored backgrounds and indicated by Roman numerals on the left. (B) Gene structures of *PoC2H2-ZF*. Exons are represented by green boxes, and introns by black lines. (C) Conserved motifs in PoC2H2-ZFPs. Five colored boxes represent the various putative motifs. The sequences of each putative motif encoded are shown in Table S2.

appeared to have high inter-clade variation (Fig. 1B). The number of introns varied from 0 to 4 in clade I, 1 to 2 in clade II, *PleosPC15_2|1104202* showed 5 exons and 4 introns, and clade IV had 1 to 7 introns (Fig. 1B).

To further investigate the structural diversity of the PoC2H2-ZFPs, the motif composition was analyzed using MEME (Fig. 1C). The results identified five putative conserved motifs (while the number was set beyond five, the *E*-value of the motif (X >5) was greater than one; data not shown). It was predicted that motif 1 encodes the conserved region ($CX_2CX_{12}HX_3H$) that corresponds to the characteristic motif of the C2H2 domain (Table S2). This motif was detected in either at the N-terminus or the C-terminus of all the PoC2H2-ZFPs (Fig. 1C). Motif 3 represented one type of C2H2 domain ($CX_4CX_{12}HX_3H$) (Table S2). PleosPC15_2|1046937, PleosPC15_2|1079678, and PleosPC15_2|1091415 in group I and PleosPC15_2|1077016, PleosPC15_2|1051309, PleosPC15_2|1054163, and PleosPC15_2|157229 in group IV possessed this sequence (Fig. 1C). It was predicted that Motif 2 encodes a false motif of the C2H2 domain lacking an H residue in the C-terminus (Table S2). It was also found in all 17 PoC2H2-ZFPs (Fig. 1C). The PoC2H2-ZFPs had diverse motif compositions. In group I, PleosPC15_2|1107157, PleosPC15_2|1088285, and PleosPC15_2|1089005 contained motifs 1 and 3, while the other genes had motifs 1, 2, and 3 (Fig. 1C). In group II, PleosPC15_2|160383 only had a single copy of motif 1, while PleosPC15_2|1111338 and PleosPC15_2|1095114 possessed motif 1 and motifs 4 and 5 in the N-terminus and the C-terminus, respectively (Fig. 1C). In group IV, PleosPC15_2|1102393, PleosPC15_2|1095325, and PleosPC15_2|1053338 shared the same motif composition, in that they all had motifs 1 and 2 in the N-terminal region. The other genes in this group had this motif composition and a single copy of motif 3 in the C-terminal region (Fig. 1C).

## Conserved domain analysis of the PoC2H2-ZFPs

To better understand the characteristics of C2H2 domains in *P. ostreatus*, multiple sequence alignment was performed to identify conserved amino acids. The results revealed that the 29 predicted C2H2 domains consisted of 23-26 amino acids (Fig. 2A). The variation in sequence length was caused by amino acid changes in two regions: $CX_{(2-4)}C$ and $HX_{(3-5)}H$. More specifically, 15 of the C2H2 domains contained two amino acids in the $CX_{(2-4)}C$ region while the others contained four amino acids (Fig. 2A). In the $HX_{(3-5)}H$ region, 24 of the C2H2 domains had three amino acids, three had four amino acids, and two had five amino acids (Fig. 2A). Two C and two H residues were conserved in every one of these domains. An F and an L residue were also highly conserved in these domains (Fig. 2A). A sequence similar to motif 1 ($CX_{(2-4)}CX_3FX_5LX_2HX_{(3-5)}H$) was also found (Fig. 2B), suggesting that motif is conserved in PoC2H2-ZFPs.

## Expression analysis of *PoC2H2-ZFs* during different developmental stages

The expression profiles of the *PoC2H2-ZFs* were measured in four different tissues (mycelia, primordia, young fruiting body, and mature fruiting body) using qRT-PCR (Fig. 3A). The results showed that *PoC2H2-ZFs* have distinctive spatial and temporal expression patterns. All the *PoC2H2-ZFs* were continuously expressed in all four tissues (Fig. 3B). *PeosPC15_2|1046937*, *PeosPC15_2|1107157*, *PeosPC15_2|1102653*, *PosPC15_2|1077016*, and *PeosPC15_2|1053338* had the lowest expression levels in all four tissues (Fig. 3B). In general, though, *PoC2H2-ZF* expression was relatively high in mycelia, primordia, and young fruiting bodies and was low in mature fruiting bodies (Fig. 3B). *PeosPC15_2|1077016*, *PeosPC15_2|112393*, and *PeosPC15_2|1088285* were highly expressed in mycelia. Five genes (*PeosPC15_2|1079678*, *PeosPC15_2|1089905*, *PeosPC15_2|1111338*, *PeosPC15_2|1095114*, and *PeosPC15_2|1095325*) were expressed more in mycelia than in primordia (Fig. 3B). The expression levels of all *PoC2H2-ZFs* increased when the primordia differentiated into fruiting bodies (Fig. 3B). As the fruiting bodies started to ripen, the expression levels of the *PoC2H2-ZFs* generally decreased. However, the expression levels of three genes (*PeosPC15_2|1088285*, *PeosPC15_2|1089905*, and *PeosPC15_2|1102653*) continued to increase (Fig. 3B).

## Expression analysis of *PoC2H2-ZFs* under auxin and cytokinin

To study the response of *PoC2H2-ZFs* to hormones, the primordia were treated with IAA and zeatin for 1 h and 3 h, respectively. Of the twelve genes studied, only the expression of *PeosPC15_2|1046937* was not affected treatment with auxin or cytokinin (Fig. 4). The expression of *PeosPC15_2|1091415* and *PeosPC15_2|1102653* was down-regulated following 1 h of treatment with zeatin, whereas the expression of *PeosPC15_2|1079678* and *PeosPC15_2|1089905* increased (Fig. 4). In addition, four genes (*PeosPC15_2|1079678*, *PeosPC15_2|1095114*, *PeosPC15_2|1051309*, and *PeosPC15_2|157229*) were up-regulated following 3 h of treatment with zeatin (Fig. 4). On the other hand, the expression of *PeosPC15_2|1079678* increased after 1 h of auxin treatment. The expression levels *PeosPC15_2|1079678*, *PeosPC15_2|1089905*, and *PeosPC15_2|1095114* increased after 3 h

A

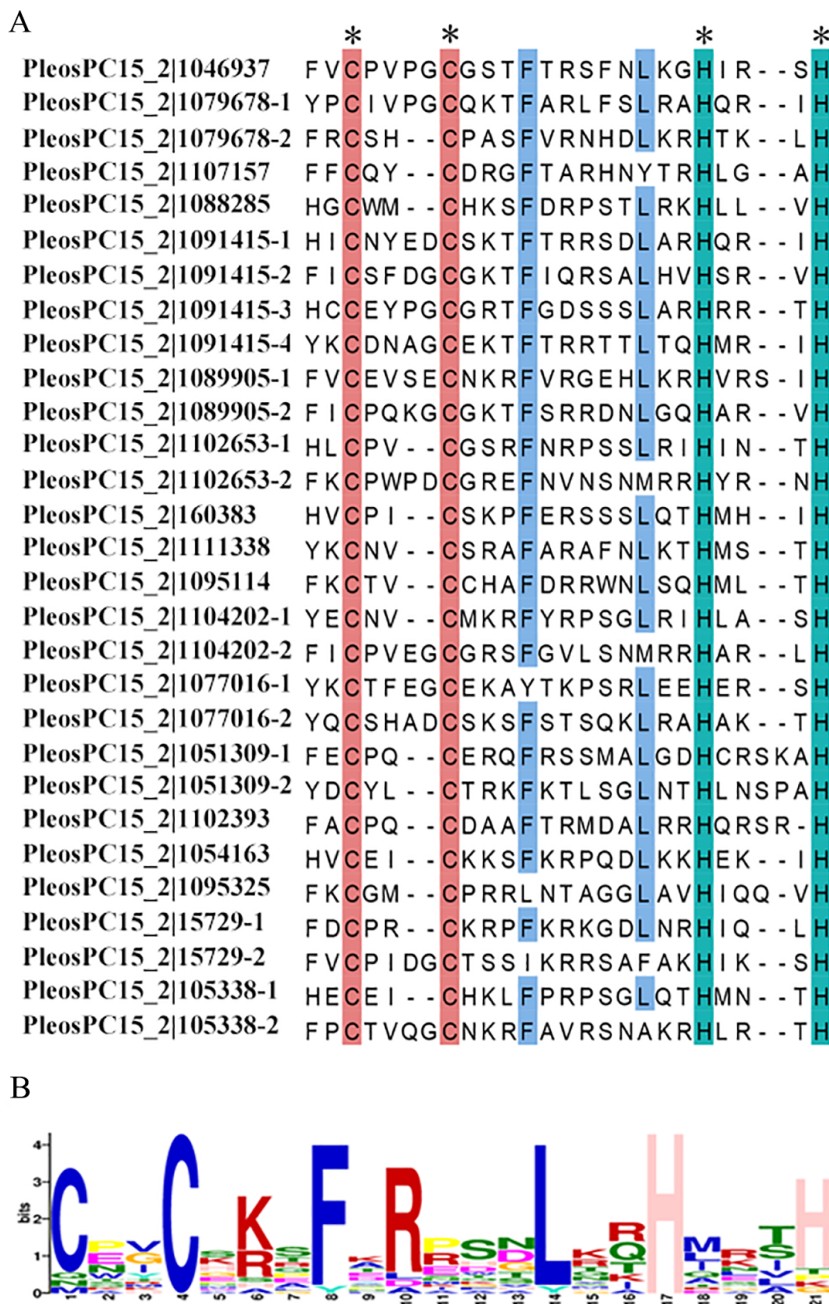

**Figure 2 Multiple alignment and conserved amino acids analysis of the C2H2 domains in PoC2H2-ZFPs.** (A) Multiple sequence alignments of the C2H2 domains in PoC2H2-ZFPs. The C2H2 domains in PoC2H2-ZFPs were predicted on SMART (http://smart.embl-heidelberg.de) (with $E$-value $< 1e^{-2}$). The conserved amino acid sequence with 100% identity was marked with an asterisk. (B) Conserved amino acid analysis of motif 1. The height of amino acids indicates the conservation ratio.

of treatment with auxin, whereas *PeosPC15_2|1102653* was down-regulated after 3 hours (Fig. 4). This shows that the *PoC2H2-ZFs* identified in this study are differentially regulated by auxin and cytokinin.

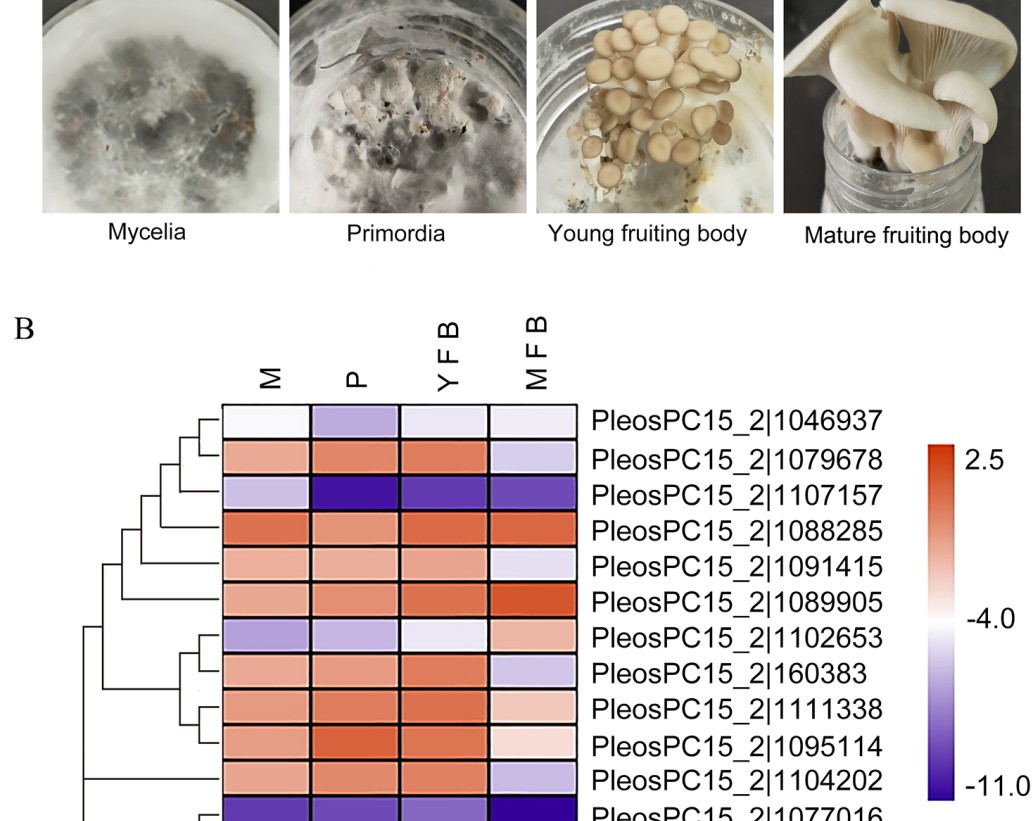

**Figure 3 Expression analysis of *PoC2H2-ZFs* in different tissues.** (A) Mycelia (M), primordia (P), young fruiting bodies (YFB), and mature fruiting bodies (MFB) were sampled to analyze the expression profiles of *PoC2H2-ZFs* in these tissues. (B) The results from RT-qPCR were log-transformed for ease of visualization in the heatmap.

## Expression analysis of *PoC2H2-ZFs* under different abiotic stresses

To investigate the potential roles of *PoC2H2-ZFs* in abiotic stress, their expression profiles were analyzed under heat and cold stress. The results showed that five genes (*PeosPC15_2|1079678*, *PeosPC15_2|1089905*, *PeosPC15_2|1095114*, *PeosPC15_2|1051309*, and *PeosPC15_2|157229*) were up-regulated after 1 h of heat stress (Fig. 5). After 3 h of heat treatment, *PeosPC15_2|1102653* was down-regulated and *PeosPC15_2|1102653* and *PeosPC15_2|1051309* were up-regulated (Fig. 5). The expression of *PeosPC15_2|1046937* and *PeosPC15_2|1102653* was significantly suppressed by 3 h of cold stress, whereas *PeosPC15_2|1111338*, *PeosPC15_2|1051309*, *PeosPC15_2|1089905*, *PeosPC15_2|157229*, and

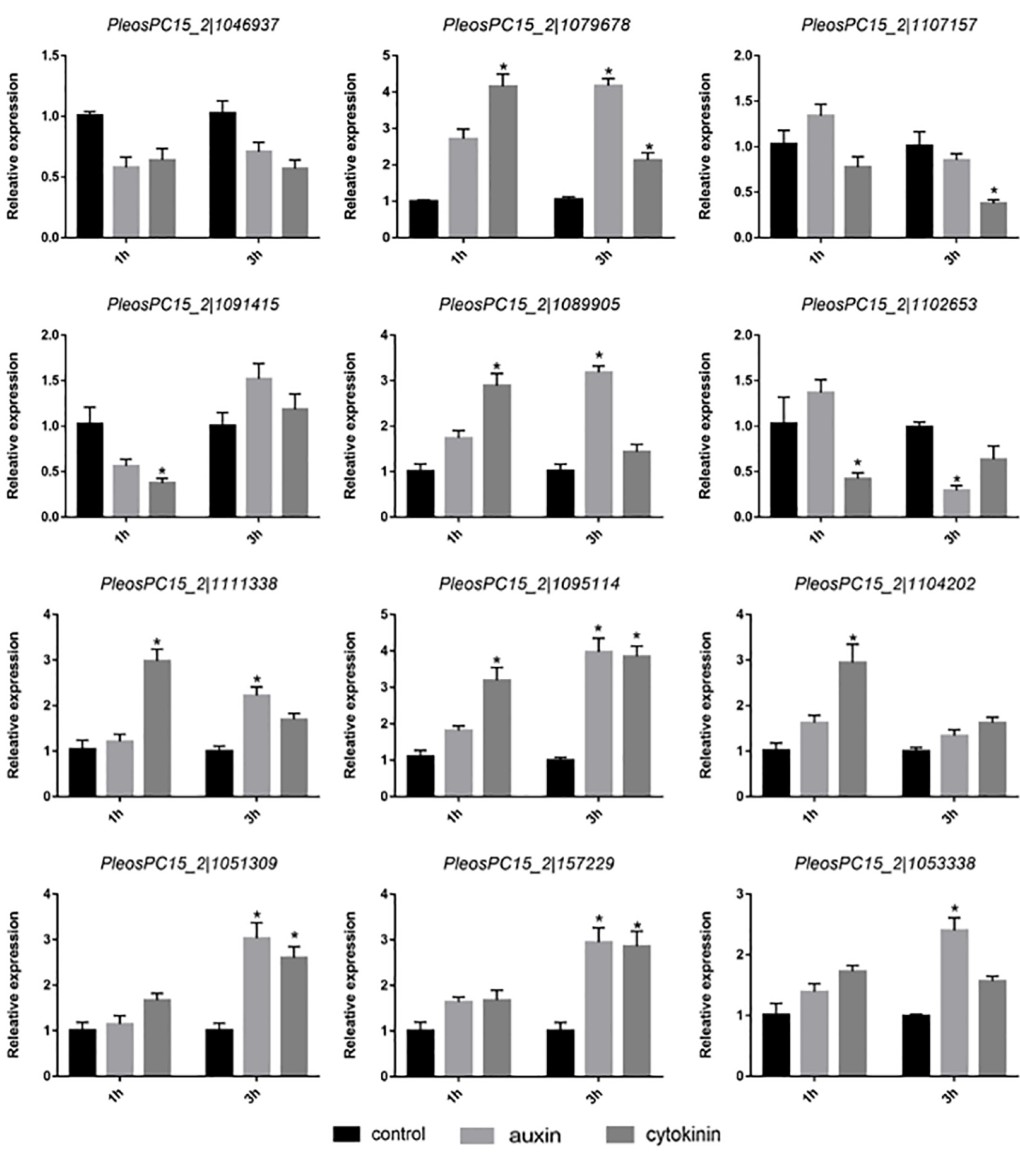

**Figure 4 Expression patterns of *PoC2H2-ZFs* under auxin and cytokinin treatment.** The level of each gene was defined as 1 in the control, and levels in IAA and zeatin treatment are presented as relative ratios. The data were analyzed using the student's *t*-test, and the asterisk indicates a significant difference at $P <$ 0.05 ($n = 3$).

*PeosPC15_2|1104202* were up-regulated (Fig. 5). The expression levels of two *PoC2H2-ZFs* (*PeosPC15_2|1091415* and *PeosPC15_2|1053338*) were not affected by cold and heat stress (Fig. 5). These data suggest that *PoC2H2-ZFs* are differentially regulated by abiotic stresses.

## DISCUSSION

C2H2-ZF proteins are one of the largest and most conserved transcription factor families in the eukaryotic kingdom. They have been reported to play important roles in mediating

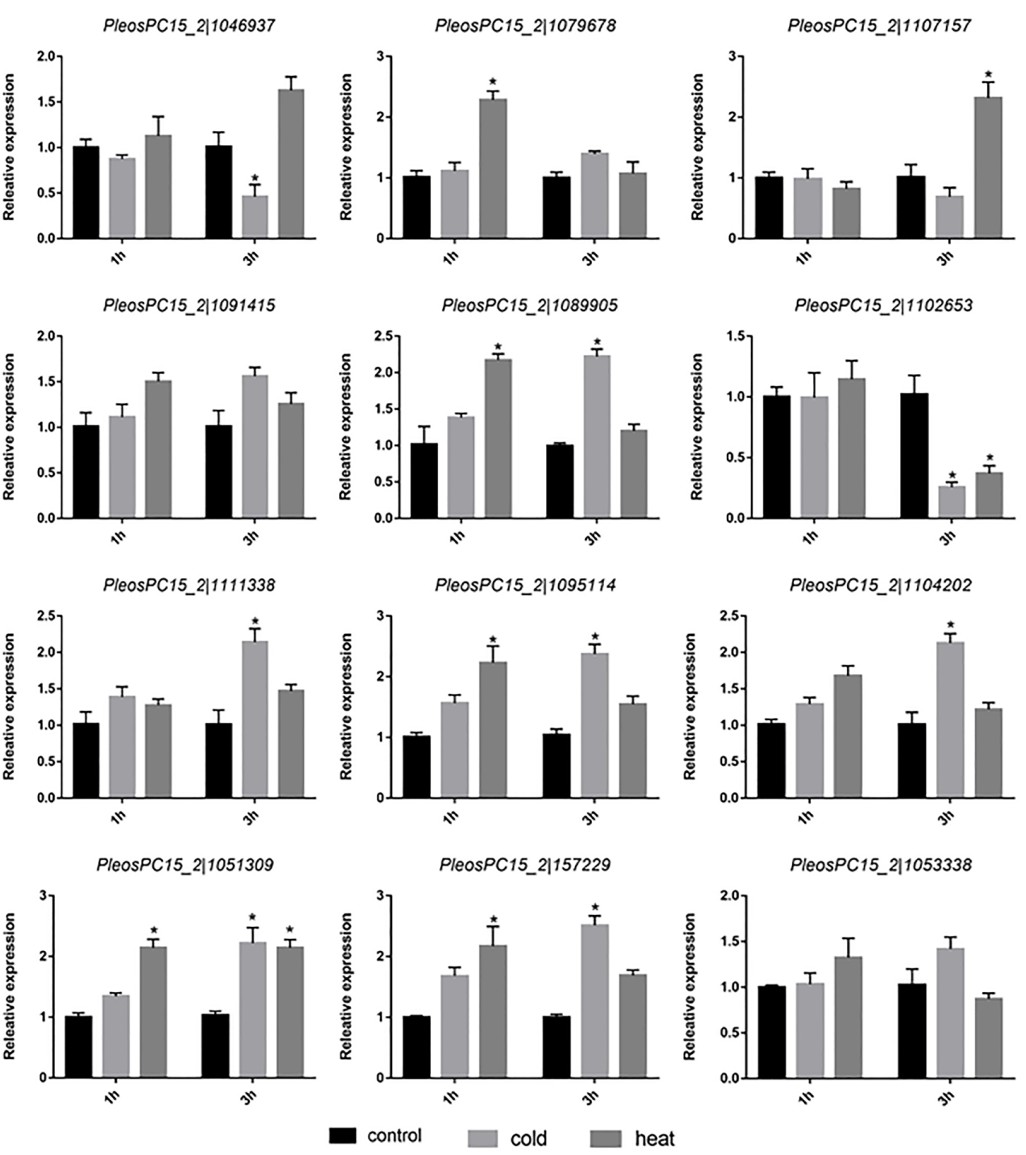

**Figure 5 Expression patterns of *PoC2H2-ZFs* response to cold and heat stress.** The level of each gene was defined as 1 in the control, and levels in cold and heat treatment are presented as relative ratios. The data were compared using the student's $t$-test, and the asterisk indicates a significant difference at $P <$ 0.05 ($n = 3$).

plant growth and responses to stress (*An et al., 2012*; *Ciftci-Yilmaz et al., 2007*; *Liu et al., 2017*; *Lu et al., 2012*; *Sun et al., 2015*; *Yin et al., 2017*). Moreover, it has been demonstrated that *C2H2-ZFs* participate in growth and development (*Tian et al., 2017*), microsclerotia formation (*Tian et al., 2017*), sexual development (*Kim et al., 2009*), and so on in fungi. For instance, it affected the yield of *Agaricus bisporus* (*Pelkmans et al., 2016*), one of the most widely cultivated commercial mushrooms. *C2H2-ZFs* could be candidate genes for edible mushrooms breeding. *P. ostreatus* is one of the widely cultivated mushrooms in China. Hence, it is useful to study *C2H2-ZFs* transcription factors in this species. The

genome of *P. ostreatus PC15* has been widely used since its release, but a systematic analysis of *C2H2-ZFs* has not yet been performed. The strain *3125* is the main cultivar for our laboratory work. However, the genome sequence of it was not obtaining. Thus, this study used the *PC15* genome to identify phylogenetic relationships, gene structures, and conserved motifs among *PoC2H2-ZFs*. In addition, the effects of growth and development, various hormones, and abiotic stress treatments on the expression of *PoC2H2-ZFs* were analyzed.

Eighteen *C2H2-ZFs* were identified in the *P. ostreatus* genome using genome-wide analysis. The proteins were further divided into four subfamilies using phylogenetic relationship analysis. However, the C2H2-ZFPs in each subfamily had low bootstrap values (Fig. 1A), which may have been caused by the low sequence similarity of the non-C2H2 domain sequences (Fig. S1). The results show that the the molecular weight, pI values, protein length, exon and intron number, and motif composition of these *PoC2H2-ZFs* (Table S1, Figs. 1B and 1C) vary widely, suggesting that *PoC2H2-ZFs* have diverse structural and physicochemical properties, as well as distinct origins and functions.

The conserved "QALGGH" sequences located in the C2H2 domain was considered the plant-specific motif that animals and yeasts lacked (*Takatsuji, 1999*)). Like the rice and tomato, most of the C2H2-ZFPs that were detected had this sequence in their genomes (*Cao et al., 2016*; Xin et al. 2019). Such sequence was not detected in the C2H2-ZFPs of *P. ostreatus* (Fig. 2A). The sequence alignment indicated that the C2H2 domains in *P. ostreatus* have a $CX_{(2-4)}CX_3FX_5LX_2HX_{(3-5)}H$ motif, and that the F and L residues are highly conserved (Fig. 2A). This signature sequence can also be written as $CX_2CX_3FX_5LX_2HX_3H$ (Table S2). This motif, called, motif 1, was detected in all PoC2H2-ZFPs (Table S2, Fig. 1C). These results suggest that the $CX_2CX_3FX_5LX_2HX_3H$ sequence was conserved in PoC2H2-ZFPs.

*C2H2-ZFs* have been shown to participate in multiple processes related to growth and development in fungi. An investigatioin of the expression profiles of *PoC2H2-ZFs* could provide information about their role in regulating the growth and development of *P. ostreatus*. In a strain of *Verticillium dahlia* with a C2H2 transcription factor loss-of-function mutation (VdMsn2), a significant reduction in hyphal growth was seen (*Tian et al., 2017*). It was hypothesized that *C2H2-ZFPs* could also influence hyphal growth in *P. ostreatus*. *PeosPC15_2|1077016*, *PeosPC15_2|112393*, and *PeosPC15_2|1088285* were highly expressed in mycelia (Fig. 3B), suggesting they play an important role in mycelial growth. Previous studies have also shown that *C2H2-ZFs* regulate the formation of primordia. Inactivation of *C2H2* in *S. commune* resulted in the formation of aggregates but not subsequent differentiation into primordia, for instance (*Ohm et al., 2011*). In this study, five genes (*PeosPC15_2|1079678*, *PeosPC15_2|1089905*, *PeosPC15_2|1111338*, *PeosPC15_2|1095114*, and *PeosPC15_2|1095325*) were expressed more in primordia than in mycelia (Fig. 3B), indicating they are involved in the formation of primordia in *P. ostreatus*. Moreover, it was shown that *C2H2-ZFs* are involved in the development of fruiting bodies. In *Aspergillus nidulans*, the deletion of *nsdC* (a gene that encoded one C2H2 transcription factor) resulted in the loss of fruiting body formation (*Kim et al. 2009*). Here, all the *PoC2H2-ZFs* was expressed more in the young fruiting bodies than in the primordia (Fig. 3B), suggesting

that *C2H2-ZFs* play an important role in fruiting body development in *P. ostreatus*. Three genes (*PeosPC15_2|1088285*, *PeosPC15_2|1089905*, and *PeosPC15_2|1102653*) showed increased expression in the mature fruiting body (Fig. 3B), as well, implying that they are involved in the ripening process of *P. ostreatus*. The *PsCZF1* gene (encoding a *C2H2-ZFs* in *Phytophthora sojae*) has been implicated in the production of oospores and swimming zoospores (*Wang et al., 2009*). Indeed, *PeosPC15_2|1088285*, *PeosPC15_2|1089905*, and *PeosPC15_2|1102653* seem to participate in spore development in *P. ostreatus* (Fig. 3B).

In *Arabidopsis thaliana*, *ZINC FINGER PROTEIN 5* (*ZFP5*) mediates the effects of cytokinin and ethylene on the formation and growth of root hairs (*An et al., 2012*). *ZFP6* has been identified as essential regulator of trichome initiation by and is responsive to gibberellin and cytokinin (*Zhou et al. 2013*). In addition, *GhWIP2* (encoded a C2H2-ZFP) mediates cell expansion during organ growth by modulating crosstalk between auxin, gibberellins, and abscisic acid in *Gerbera hybrida* (*Ren et al., 2018*). Thus, it was hypothesized that various hormones can regulate *C2H2-ZFs* which can, in turn, affect developmental processes. *P. ostreatus* produces auxin (*Bose, Shah & Keharia, 2013*), and exogenous auxin and cytokinin have been reported to affect mycelial growth (*Ramachela & Sihlangu, 2016*). In this study, expression levels of *PoC2H2-ZFs* in primordia changed significantly in the presence of auxin and cytokinin (Fig. 4). Therefore, auxin and cytokinin have the potential to affect *PoC2H2-ZF*-mediated growth and developmental processes.

Previous studies have revealed that *C2H2-ZFs* confer resistance to abiotic stress in plants. In *Arabidopsis*, root growth in transgenic plants constitutively expressing *Zat10* were more tolerant to heat stress (*Mittler et al., 2006*)). Overexpression of *ZAT18* in transgenic *Arabidopsis* plants also increased drought tolerance, whereas the mutation of this gene resulted in decreased drought tolerance (*Yin et al., 2017*). In soybean, the expression of the *C2H2-ZF* gene *GmSCOF-1* was induced by low temperature, and the overexpression lines not only had increased cold tolerance, but also had increased expression levels of cold-responsive genes (*Kim et al. 2001*). During *P. ostreatus* cultivation, bad environmental conditions negatively impact the growth and development of the mushrooms by inhibiting mycelial growth and disrupting the integrity of the cell wall, thereby increasing the risk of fungal contamination and reducing yield (*Qiu et al., 2018*). In this study, *PoC2H2-ZFs* were induced by heat and cold stress (Fig. 5), meaning that this transcription factor may have a conserved function related to heat and cold tolerance in fungi. Notably, *PeosPC15_2|1089905* and *PeosPC15_2|1102653* were induced by heat and cold stress (Fig. 5), suggesting that these genes play a variety of roles in response to various stresses.

## CONCLUSIONS

In this study, we identified 18 *C2H2-ZFs* in the *P. ostreatus* genome. Their phylogenetic relationship, gene structure, motif composition, and other structural factors were highly variable. The expression profiles of these *PoC2H2-ZFs* suggest they play diverse roles in tissue growth and development. In addition, hormones and abiotic stress treatments induced the expression of these *PoC2H2-ZFs*, meaning that they could also participate in hormone signaling and abiotic stress response pathways.

## ACKNOWLEDGEMENTS

The authors sincerely thank the anonymous reviewers for their affirmation and valuable comments.

### Funding

This work was supported by the Technology System of Vegetable industry in Anhui Province (Edible Mushrooms) (No.AHCYJSTX-09-5) and, the Research Projects of the Anhui Academy of Agricultural Sciences (No.2020YL028). The funders had no role in study design, data collection and analysis, decision to publish, or preparation of the manuscript.

### Grant Disclosures

The following grant information was disclosed by the authors:
The Technology System of Vegetable industry in Anhui Province (Edible Mushrooms): No.AHCYJSTX-09-5.
The Research Projects of the Anhui Academy of Agricultural Sciences: No.2020YL028.

### Competing Interests

The authors declare there are no competing interests.

### Author Contributions

- Qiangqiang Ding conceived and designed the experiments, performed the experiments, analyzed the data, prepared figures and/or tables, and approved the final draft.
- Hongyuan Zhao and Xiangting Jiang performed the experiments, prepared figures and/or tables, and approved the final draft.
- Peilei Zhu performed the experiments, analyzed the data, prepared figures and/or tables, and approved the final draft.
- Fan Nie and Guoqing Li conceived and designed the experiments, authored or reviewed drafts of the paper, and approved the final draft.

### Data Availability

The raw data is available in the Supplemental File.

### Supplemental Information

Supplemental information for this article can be found online at http://dx.doi.org/10.7717/peerj.12654#supplemental-information.

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
