# Peer review of "Genome-wide identification and expression analyses of C2H2 zinc finger transcription factors in Pleurotus ostreatus"

_PeerJ, doi:10.7717/peerj.12654_

## Round 0.1 · original submission · Major Revisions

Your manuscript has been reviewed by four reviewers. Two reviewers suggested major revisions and two reviewers suggested minor revisions.

We hope that their comments prove useful to you as a guide for revisions prior to resubmission.

We ask that you resubmit a tracked copy showing all edits, a clean revised copy with all changes accepted, and a point by point response to the reviews. We anticipate that revisions may require 4-8 weeks. If you require more time, please let us know and an extension can be granted.

We look forward to receiving your revised version.

·

Basic reporting

This MS aims to uncover Genome-wide characterization of the C2H2 zinc finger transcription factors in Pleurotus ostreatus and expression analyses under abiotic stress. In my opinion, it meets the journal's basic requirements.

Experimental design

It would be better to explan the basis of setting heat and cold stress treatment conditions in Introduction. the primordia were cultured at 38 °C for 1 h 3h for heat stress treatment, it is somewhat incomprehensible.

Validity of the findings

All the bioinformation was infered from PC15. While P. ostreatus 3125 was used in this study. There are not the same strain. It would be better to be discussed in the discussion.

Additional comments

Line 118, "Scientific Edible Fungal" ?

Reviewer 2 ·

Basic reporting

In this study, author used bioinformatics method to analyze the C2H2-ZFPs of Pleurotus ostreatus. 18 C2H2-ZFs were identified in P. ostreatus genome. The methods and results are acceptable. However, I think your research is too simplistic. I hope you can add some analysis and experiments. As far as writing, there are lots of small problems should be fixed.

Experimental design

no comment

Validity of the findings

no comment

Annotated reviews are not available for download in order to protect the identity of reviewers who chose to remain anonymous.

Reviewer 3 ·

Basic reporting

The English language should be improved to ensure that an international audience can clearly understand your text. Dozens of grammar mistakes in the manuscript should be corrected. Such as Line74-75, 127, 157, 168, 188,190,192, 218 and so on.

Experimental design

no comment

Validity of the findings

no comment

Additional comments

I suggest the authors cluster the genes in the heatmap of Fig.3b to make it more visual and helpful for analysis of the similarity of different genes.

Reviewer 4 ·

Basic reporting

Dear Authors,

Thank you for the opportunity to review your manuscript. It is solid and reliable work and I believe it is worth to be published on the pages of PeerJ journal. However, some changes need to be done. First of all, the title doesn't correspond to the content of your article. Basically, you discuss the influence of abiotic (temperature) and biotic (developmental stages, hormones) on the expression of the studied genes. Second, the English language should be considerably improved. I suggest that the manuscript should be corrected by the English native speaker and researcher. I have some concerns regarding the figure legends, namely figures 4 and 5. They reflect the methods used. I would prefer to see the actual finding: visually in your plots and written in the legends. I would also recommend using keywords different from the title, which might increase the "visibility" of your paper for the web search engines, but this is rather a recommendation, not mandatory.

Respectfully,
Reviewer.

Experimental design

well done

Validity of the findings

no comments

---

## Round 0.2 · accepted · Accept

Thank you for revising the manuscript in response to reviews.